# LEMON: A Lightweight Facial Emotion Recognition System for Assistive Robotics Based on Dilated Residual Convolutional Neural Networks

**DOI:** 10.3390/s22093366

**Published:** 2022-04-28

**Authors:** Rami Reddy Devaram, Gloria Beraldo, Riccardo De Benedictis, Misael Mongiovì, Amedeo Cesta

**Affiliations:** 1CNR—Italian National Research Council, Institute of Cognitive Sciences and Technologies, Via Gaifami 18, 95126 Catania, Italy; misael.mongiovi@istc.cnr.it; 2CNR—Italian National Research Council, Institute of Cognitive Sciences and Technologies, Via S. Martino della Battaglia 44, 00185 Rome, Italy; gloria.beraldo@istc.cnr.it (G.B.); riccardo.debenedictis@istc.cnr.it (R.D.B.); amedeo.cesta@istc.cnr.it (A.C.)

**Keywords:** emotion recognition, face recognition, assistive robotics, deep convolutional neural networks, computer vision

## Abstract

The development of a Social Intelligence System based on artificial intelligence is one of the cutting edge technologies in Assistive Robotics. Such systems need to create an empathic interaction with the users; therefore, it os required to include an Emotion Recognition (ER) framework which has to run, in near real-time, together with several other intelligent services. Most of the low-cost commercial robots, however, although more accessible by users and healthcare facilities, have to balance costs and effectiveness, resulting in under-performing hardware in terms of memory and processing unit. This aspect makes the design of the systems challenging, requiring a trade-off between the accuracy and the complexity of the adopted models. This paper proposes a compact and robust service for Assistive Robotics, called *Lightweight EMotion recognitiON* (LEMON), which uses image processing, Computer Vision and Deep Learning (DL) algorithms to recognize facial expressions. Specifically, the proposed DL model is based on *Residual Convolutional Neural Networks* with the combination of *Dilated* and *Standard Convolution Layers*. The first remarkable result is the few numbers (i.e., 1.6 Million) of parameters characterizing our model. In addition, *Dilated Convolutions* expand receptive fields exponentially with preserving resolution, less computation and memory cost to recognize the distinction among facial expressions by capturing the displacement of the pixels. Finally, to reduce the dying ReLU problem and improve the stability of the model, we apply an *Exponential Linear Unit* (ELU) activation function in the initial layers of the model. We have performed training and evaluation (via one- and five-fold cross validation) of the model with five datasets available in the community and one mixed dataset created by taking samples from all of them. With respect to the other approaches, our model achieves comparable results with a significant reduction in terms of the number of parameters.

## 1. Introduction

Mobile Telepresence Robots represent a class of robotic platforms, characterized by a video conferencing system mounted on a mobile robotic base, which allows a remote user to move around in the robot’s environment. Although the technology used on these robotic platforms has evolved considerably in recent years [1], such tools are basically relegated to provide telepresence services on different mobile robotic platforms that can be controlled remotely [2,3,4], having no or minimal autonomy [5]. Being relatively inexpensive, these platforms can be used extensively, for example, to mitigate the isolation of elderly people living alone [6,7]. In order to become user-effective, however, it has arisen the needs of integrating the robotic progress with additional technology, so as to create an end-to-end system that can effectively help to improve the quality of life. The low performances of hardware in such platforms, however, makes these tools unsuitable for supporting particularly complex data processing such as the trendy neural models.

The use of deep neural networks, specifically, has become popular in the last years, enhancing the functioning of wearable sensors and smartphones but also in the niche of voice assistants and robotic platforms. Deep Learning models, however, usually need devices with sizeable computational power. With the increase in performance requirements, in fact, the number of layers and, with them, the number of model parameters, is growing more and more, raising problems related to performance, both in terms of prediction efficiency and in terms of the space required for storing the models [8,9]. Smaller models (e.g., less parameters), on the other hand, require less computation to predict their results. Compared to larger models, indeed, smaller models run faster and require less memory. This makes them more suitable to be executed on devices that can rely on slow processors and low system memory.

In this work, we propose a real-time intelligent service called *LEMON* (from *Lightweight EMotion recognitiON*) that relies on *Residual Dilated Convolutional Neural Networks* (RDCNN) for continuously monitoring users’ emotional status by analyzing their facial expressions. LEMON’s lightweight architecture permits running the service even on not particularly well-performing hardware, such as those of commercial telepresence robots, allowing it to adapt the system’s behavior to the perceived emotions and, consequently, to adopt a more empathic behavior towards the user.

The paper is structured as follows. Section 2 introduces some of the related works about emotion recognition. Section 3 presents some technical background required for understanding the architecture proposed in Section 4. Section 5 describes the adopted platform and the databases exploited for training the proposed model. Section 6 describes the experimental procedure and compares the results with existing works. Finally, Section 7 closes the paper.

## 2. Related Works

Emotions represent individuals’ feelings which are often communicated through expressions rather than vocally or verbally. Facial Expressions are the key points to recognize emotions and are used to analyze non-verbal communications. Understanding facial expression is an essential part of nonverbal communication. Ignoring the expressions of an interlocutor, for example, could easily lead to the loss of information indeed. While the face betrays the actual feeling, words may not match a person’s emotions. Previous research reveals two kinds of human emotions: *basic* [10] and *complex* [11]. The primary emotions are categorized as *Anger*, *Disgust*, *Fear*, *Happiness*, *Sadness*, and *Surprise*. Such emotions can be easily recognized through changes in facial expressions, conveyed through a combination of gestures involving, for example, the eyes, eyebrows, lips/mouth, nose and cheeks. Complex emotions, on the contrary, are categorized as *Frustration*, *Confusion*, *Boredom*, *Flow* and *Delightful*. By involving the surrounding environment and, hence, a context rather than just the people’s expressions, these emotions, in general, are harder to be recognized.

Every gesture also reveals a facial expression. For example, as regards mouth gestures, while an open mouth might indicate fear, raising the corners of the mouth might represent happiness. Lowering the corners of the mouth might convey sadness while the biting the lips might indicate anxiety. Although humans are very good at recognizing emotions, machines must recognize facial expressions starting from facial gestures called facial features. Such features, additionally, are usually stored on datasets which, depending on how images were taken, might present significant differences related, for example, to lighting, posture, face occlusion, skin color, etc. [12]. With this purpose, traditional Machine Learning (ML) algorithms have applied specific hand-crafted features [13,14]. However, such methodologies suffer from many limitations, including the difficulty of generalizing an effective emotional model which performs in real-time. Moreover, the model has to obtain a specific translation, rotation and distortion invariance to understand the subject of interest within the image. Every image holds a particular spatial connection, neural networks can observe the local pixels and then unite the local information to perceive the global information at a high level and it needs to expand the network to learn, based on the complexity of the data. Nevertheless, previous research reveals that using the combination of advanced computer vision and deep learning techniques allows achieving optimal real-time performance by overcome challenges rising from data variability [12,13,14,15,16,17].

*Deep Convolutional Neural Networks* (DCNN), in particular, have a a powerful ability to extract discriminant features automatically and have been used for various recognition applications for decades. In the research community, many Deep Neural Network architectures are proposed to solve various complex problems in different fields, including emotion recognition using *VGG* [18], *ResNet* [19] and *AlexNet* [20]. However, those deep networks have millions of floating points (e.g., trainable parameters/weights) that need to be computed, requiring substantial computational power. The above networks are hardly trainable from small datasets, leading to underfitting the model while generalizing, hence diminishing the performance. Furthermore, they are not suitable for devices with limited hardware configuration.

However, to optimize or modify the exiting state of the art models according to the problem, Yanling et al. [21] modified VGG16 used as a part of their model, Yizhen and Haifeng [22] modified *ResNet* and Devaram et al. [23] modified Mobilenet to develop facial identification system for small memory microcontrollers. There is the need to optimize the network structures according to available data and the specific applications.

Current research is focused on developing smaller models for low-cost real-time systems. Lee et al. [24] proposed a nano deep learning model for emotion recognition to perform in small memory devices run in real-time, and Kuang et al. [25] used tried to avoid train large DL models to recognize fashion images using mobile devices. In recent advancements, mobile devices have gained sufficient memory, yet mobile devices have to run many other applications in parallel with in the devices. For this reason, Zhao et al. [26] developed a facial age estimation recognition system through tiny models to consume low memory in real-time. Deep neural networks show great performance improvements by using residual connections instead of plain feed-forward networks [27], and Devaram et al.’s work [28] shows both dilation and standard convolutional neural networks used to classify hyperspectral images, which proves that such combinations stabilize the network over different datasets.

## 3. Technical Background

This section briefly describes basic concepts concerning state-of-the-art methods for Face Detection, including the management of Data Imbalance and common techniques to extract and learn deep discriminate features. We build upon these concepts in designing our architecture. Further details about the modifications of our architecture with regard to the state-of-the-art are presented in Section 4.

### 3.1. Face Extraction

Face detection is the initial step for the emotion recognition system in the pre-processing stage. Indeed, in real-time applications, cameras capture not only the people’s face but also the surroundings, introducing noise. With this aim, specifically, our model relies on one of the accurate and fastest classifiers, the *Viola-Jones Face Detection* Technique [29], also called the *Haar Cascade classifier*. It detects the regions of the images corresponding to face and crop the background.

### 3.2. Data Imbalance

Data Imbalance, namely, the differences in terms of number of samples per each class, represents one of the limitations of the current machine learning approaches. Sampling techniques can be used to solve the data imbalance problem, such as *Undersampling* and *Oversampling* [30]. *Undersampling* reduces the number of samples in the majority class, while *Oversampling* increases the sample to equal the majority class. The datasets we used for creating the proposed model, described in the previous section, have very little data, and they are substantially imbalanced, which leads to underfitting. To overcome this limitation, we have applied oversampling techniques after dividing the data into train, validation and test sets.

### 3.3. Residual Learning

The main intuition behind the use of a residual architecture is that traditional (plain) deep neural networks’ ability to learn low-, medium- and high-level features increases as the number of layers (depth) increases, leading to better accuracy. However, if the depth of the network increases excessively by stacking more and more layers, this leads to higher training errors, and the accuracy becomes saturated.

Hence, He et al. [27] proposed a Residual Learning architecture, which simplifies the architecture of traditional Convolutional Neural Network (CNN) by adding a parallel operation over (two) hidden layers of the network, called a “skip connection”, shown in Figure 1. The figure represents a residual building block, where *X* is an input, and F(X) is a non-linear convolution operation over hidden convolutional layers (*weight layers*). Hence, in residual learning, the presence of a shortcut/skip connection applies in parallel over two convolutional layers (*weight layers*), which applies an identity mapping over its input *X*. The output of both branches are then summed up.

When a non-linear convolution operation is applied over an input image *I* with size N×N and a convolution filter *F* having size M×M, the output is a feature map of size $O_{I}$ given by:(1)$$O_{I}=\frac{N-M+2\cdot P}{S}+1$$
where *P* represents the padding (number of pixels on each side) and *S* the stride.

Note that the skip connection does not need any additional parameter to increase performances. Therefore, we can reduce the number of layers of the network while preserving performances, leading to a significant parameter reduction.

### 3.4. Dilated Convolutions

In literature, a Dilated Convolution (*D-Conv*) [31] or *Atrous Convolution*, represents a convolution operation that supports the exponential expansion of the receptive field, i.e., the active region of the convolution operation over input increases exponentially, without increasing the number of trainable parameters. Furthermore, padding is performed to preserve the data dimensions at the output layer. As a result, *D-Conv* obtains more global information while capturing the local context, and padding does not drop any parameters in the feature map. Moreover, *D-Conv* helps extract more contextual information than standard discrete convolution to improve the performance. The architecture of *D-Conv* involves several layers, each of them computing the following output:(2)$$O\left(i,j\right)=\sum_{l=1}^{X}\sum_{w=1}^{Y}I\left(i+dr*l,j+dr*w\right)\cdot k\left(l,w\right)$$
where *I* is the input image (single channel feature map), *O* is the output (single channel feature map) and k(l,w) is the value of the filter matrix in row *l*, column *w*. The architecture has many layers with exponentially increasing dr. The size of the receptive fields varies with the selected dilation rate dr; if dr=1, the output of the resulted feature maps is the same as for the standard convolution layer. Yu et al. [31] also proposed to remove the max-pooling layer to better capture the information. As we discuss below (Section 4), our neural network model partially incorporates a *D-Conv* layer in the convolution and in the identity block of the Residual architecture.

### 3.5. Activation Functions

In Artificial Neural Networks, the activation function of a node represents the output of that node given an input or set of inputs. Surely, every activation function has both advantages and drawbacks, making the choice challenging. For instance, the *Rectified Linear Unit* (*ReLU*) one of the most widely used non-saturated activation functions, improves the computation speed. On the other hand, *Exponential Linear Unit* (*ELU*), introduced in 2016 by Djork-Arné Clevert [32], is computationally intensive because it can fire at negative inputs, resulting in exponentially increasing weights.

Moreover, *ReLU* suffers from the vanishing gradient (or dying neuron) problem when negative input occurs, while *ELU* does not have the vanishing gradient effect.

It is also challenging to generate non-linearity with piece-wise linearity and vice versa, and efficiently estimate the smooth polynomial functions with *ReLU*, while *ELU* is capable of dealing with linearity and non-linearity simultaneously. Therefore, to prevent the information loss (vanishing gradient) and fast computation, we used both *ELU* and *ReLU* activation functions in the same network: *ELU* is applied only in the first layer of each stage, and the rest of the network uses the *ReLU* activation function. This setting produces optimal and stable performances.

## 4. The Lightweight EMotion recognitiON System

In this work, we propose a *Residual Dilated Convolutional Neural Network Architecture*, we named *LEMON* (*Lightweight EMotion recognitiON system*), which uses global and local features to classify emotions from facial expression in real-time. The entire pipeline is shown in Figure 2. We build upon the work of He et at. [27]. Residual learning proves that the a deeper model with less parameters can achieve greater performances. Given such advantages (as discussed in Section 3.3), we employed the residual architecture scheme as shown in Figure 2. The proposed residual architecture consists of three main blocks: input, residual and output. The residual block mainly consists of two shortcuts (skips connections) modules. The first module is a convolution block with a 2D convolution layer as a shortcut, which makes the output size smaller than the input. The other module is an identity block that does not contain any convolution layer as shortcut and produces an output with the same size as the input.

The architecture relies on the following sub-model components: *2D-Convolution* and *Dilation Layer*, *ELU* and *ReLU* activation functions, *Configuration of Convolution Filters*, *Regularization*, *Batch Normalization* and *AveragePooling*. *LEMON* is designed to reduce the computation cost and stabilize the network with a small number of training samples by stacking shortcut/skip connections with the existing feed-forward network. *Residual Learning* generalizes the deeper model with a smaller number of trainable parameters without compromising the performance. The residual block, in particular, allows the stacked layers to fit the residual mapping better than directly. Moreover, the *Dilated Convolution Layers* expand the receptive field, which brings higher performances. In order to expand the receptive fields, we might have used a standard CNN with pooling layers, which reduces the feature maps, and a subsequent upsampling layer to increase the feature map size. However, the sequence of such reduction and expansion procedures would have lead to the loss of essential information. Instead, *Dilated Convolutional (D-Conv) Layers* expand the receptive fields exponentially, carrying more information through sparse feature maps.

It is worth noticing that, as already mentioned in Section 3.5, one of the most significant activation functions, the *ReLU*, by allowing only positive inputs during backpropagation, suffers from the vanishing gradient problems. To address this issue, we adopt an *ELU* activation function, which improves classification accuracy and the stability of the model. It is worth noticing that, in our approach, *ELU* and *ReLU* work together. In particular, *ELU* is used only in the initial layer of each Convolution and Identity block to prevent from vanishing gradient without compromising computation speed. Previous research, indeed, proved that *ELU* enhances the performance on unseen data. Devaram et al. [28] employed ELU to stabilize the network in classifying various hyper-spectral images having different spectral and spacial information for remote sensing, and Devi et al. [33] have conducted experiments on Natural Language Processing on tasks such as sentiment analysis with *ELU* and *ReLU*, showing that the use of *ELU* produced better performances than the unique *ReLU* activation function with different input types.

The *LEMON* architecture, represented in Figure 2, aims to find the correspondence between the input (image) and the output (predicted category) by taking advantage of skip connections, instead of directly mapping from the input to the actual output. Specifically, the architecture consists of eight building blocks: one input block, containing a 2D Convolution, a Batch-Normalization, an ELU activation function, a Max_Pooling and a Dropout layer, followed by six residual blocks. Finally, the output block contains an Average_pooling layer, a Dense layer, a Fully connected layer and a Softmax layer. The proposed residual blocks, furthermore, are composed by two parts: a *convolution* block and an *identity* block. The *convolution block* includes five convolution operations, while the *identity block* contains four convolution operations in each stage. The proposed LEMON architecture contains in total 55 convolutional layers. Note that the number of filters to increase the depth of the feature maps are incremented by 16 each stage. Such a filter configuration (Figure 3) leads to an efficient model with a smaller number of trainable parameters.

In detail, the input block has a convolution layer with 32 kernels of size 3×3 and stride 2, and it is fed with the preprocessed grayscale images with 120×120 as their size. The initial layers aim to extract edges, oriented-edges, corners, and shape features. Then, *Batch Normalization* is applied on the output feature map of the first layer. The output is passed to the *max-pooling layer*. The *Max-pooling layer* is analyzed to locate the underlying features of an image in each dimension. Then, the first layer’s result is transferred to the residual block. The identity block passes its feature maps by skipping four convolutional layers without additional processing before the summation function. Skip connections are standard modules in various deep neural network architectures. They provide an alternative path for the gradient, which is often beneficial for the model convergence by skipping some layers in the neural network. Taking inspiration from [27], in which the skip connection in the identity block provides an identity mapping, as shown in Figure 2, we combine the information from $x_{c}$ and $f_{3}\left(x_{c}\right)$ and pass it to the subsequent layer.

Conversely, we modified the original ResNet module, in which the skip connection in the identity block is applied over two blocks. In the proposed architecture we have large feature maps generated from the dilation layers. We then need some more convolution operations to reduce the dimension of the feature map; therefore, we apply a longer skip connection that “skips over four blocks”. The advantage is to detect small or fine-grained details from layers that are closer to the input. This upgrade showed a significant performance improvement.

Another important point is that the convolution block appears at the initial stage of every significant residual block. As a result, the size of the input feature maps is reduced due to kernel stride, and the number of feature maps is increased due to increasing convolutional kernels. At the same time, the identity block applies identity mapping over the data coming from the convolutional block. We assigned the exact number of filters to both convolution and identity blocks. The complete filter configuration is shown in Figure 3.

Finally, the *AveragePooling layer* allows the model to process the features from the previous residual block without reassigning the number of connections in the fully connected (FC) layer. The final layer of LEMON is an output softmax layer with the output nodes corresponding to the emotion categories we need to classify.

## 5. Materials and Methods

### 5.1. Commercial Robot

This section presents the commercial robot adopted as a target to run the framework. Secondly, we provide the description of the datasets available in the literature for creating and validating the proposed model. Finally, we briefly introduce some state-of-the-art Deep Learning datasets as support for understanding the pipeline behind the proposed model.

The ultimate goal of LEMON is to recognize human emotions on tools with poorly performing hardware. To this end, we adopted an Ohmni robot as our commercial robotic platform to finally test the proposed model. The robot, in particular, was constituted by a differential mobile platform endowed with two 2D RGB cameras: Frontal (Supercam FOV), placed on the top of the robot’s tablet, and Navigation, under the robot’s neck in Figure 4.

For the emotion recognition purpose, we used the Frontal camera, which has the following specifications: 2 MP as resolution, 3.0 micron as pixel dimension and 5865 × 3276 μm as the sensor’s dimensions. We acquired and processed the incoming images inside the Robot Operating System (ROS), the standard de facto in Robotics (Specifically, we used the package *usb_cam* http://wiki.ros.org/usb_cam, accessed on 20 March 2022), which we integrated in the robot by exploiting *Docker* virtualization. The input images for testing the model were characterized by a 640 × 480 resolution and *RGB8* as the encoding. As we mentioned before, this robotic platform has limited computational resources, representing one of the technological challenge of this work. Specifically, it has Intel Atom X5 Z8350 as processor with 1.92 GHz, with 4 cores, DDR3L memory with 2 GB and without a GPU.

### 5.2. Datasets

To train and evaluate the proposed model, we exploited five state-of-the-art datasets which considered seven to eight basic emotions (e.g., anger, (contempt), disgust, fear, happiness, naturalness, sadness, surprise). All the datasets are publicly available for the research community. These datasets have different characteristics in terms of resolution, scale, gender, age, race, etc. The technical details of the dataset are reported in the following subsections. In addition, we created one new dataset, which we called Mixed, including samples from the previous ones.

#### 5.2.1. The Extended Cohn-Kanade (CK+)

The Extended Cohn-Kanade dataset (CK+) [34,35] is one of the benchmark public datasets regarding action units and facial emotions of people aged 18 to 50 years old. The dataset collects the facial behavior of 210 adults using 2 synchronized Panasonic AG-7500 cameras. The data are recorded as a sequence of images in 30-degree frontal views with 640 × 480 pixels Figure 5a. The recorded sequence of images begins with the neutral expression, and ends with the related emotion. Specifically, the dataset considers the following primary facial expressions: Anger, Disgust, Fear, Happiness, Naturalness, Sadness and Surprise.

#### 5.2.2. The Japanese Female Facial Expression

The Japanese Female Facial Expression (JAFFE) database [36] contains 213 frontal view images of 10 different expressions from 10 Japanese females. For each person, the dataset has 3–4 samples with basic facial expressions: Anger, Disgust, Fear, Happiness, Sadness and Surprise, including Natural expressions. Examples from JAFFE datasets are shown in Figure 5b, and each image has 256 × 256 pixels.

#### 5.2.3. The Karolinska Directed Emotional Faces

The Karolinska Directed Emotional Faces (KDEF) dataset contains 4900 images with 6 different emotional expressions (anger, disgust, fear, happiness, sadness, naturalness and surprise) from 5 different angles in Figure 5c. It includes images of 70 individuals (35 females and 35 males) adults between 20 and 30 years old. Each image was taken without occlusions, such as mustaches, earrings, eyeglasses, etc.

#### 5.2.4. Taiwanese Facial Expression Image Dataset

The TFEID dataset [37] contains 7200 stimuli captured from 40 Taiwanese models (50% male) aged between 18 and 30 years old, under different angles (0 and 45), with both high and slight intensities. The dataset contains 259 images of size 480 × 600 pixels with the following emotions: 31 samples for anger, 47 contempt, 36 disgust, 22 fear, 31 happiness, 32 neutral, 30 sadness and 20 surprises.

#### 5.2.5. Sase-Fe

The SASE-FE dataset [38] includes 643 videos of 54 participants between 19 and 36 recorded by a high-resolution GoProHero camera with 100 frames per second, and it was about 3 to 4 s. The videos represent six universal expressions (e.g., Anger, Happiness, Sadness, Disgust, Contempt, and Surprise) for reliable and fake expression. For our purpose, we have used only the samples labeled as reliable expressions. Some examples are shown in Figure 5d.

In each video, subjects started from a neutral emotion (e.g., the length of this neutral emotion is not predefined), reproduced an emotion and returned to the natural state.

#### 5.2.6. Mixed Dataset

Mixed Dataset have been achieved by combining all the previous ones. Since the difference in the considered emotions among the datasets (e.g., seven vs. eight); first, we have merged the data in each category. Thus, the mixed dataset considers eight emotions (Anger, Contempt, Disgust, Fear, Happiness, Naturalness, Sadness and Surprise). In total, it has 11,076 samples.

## 6. Experimental Procedure and Results

This section presents the description of the proposed training and evaluation pipeline in Figure 6, which includes the Data Preprocessing, partitioning and augmentation operations. Then, details about the experimental setup have been reported. Finally, we show the comprehensive performances, and we compare them with some state-of-the-art approaches.

### 6.1. Preprocessing

In the initial phase of preprocessing, the RGB images are converted to Grayscale. Then, we extract the faces using the Viola-Jones Face Detection Model (see Section 3.1). All the cropped faces are zero-centered normalized, consisting of global contrast normalization (GCN) and local normalization. Since the examined datasets contain images by different resolutions, we train the CNN with a unique resolution, 120 × 120 pixels, respectively.

Finally, the preprocessed dataset was randomly shuffled and split into 70%, 20% and 10% as train, validation and test sets, respectively. Furthermore, the oversampling technique in Section 3.2 is applied on both training and validation sets to distribute the data into an equal number of samples in each category of the dataset. The resulting dataset is passed in input to the LEMON Architecture.

### 6.2. Experimental Setup

The entire framework, including preprocessing, training and evaluation, was developed using Python programming, OpenCV, Keras as backend API within the Ubuntu-18.04 operating system. We trained the network from scratch on NVIDIA Quadro RTX 6000 GPU having memory of 24 GB RAM.

Hyper-parameter selection for both network and learning is also essential. Initially, Network weights are initialized with random Gaussian zero mean and 0.05 standard deviation. The batch size is 10 samples, and early stopping is set between 200 and 500 epochs; this value depends on the number of training samples in the examined datasets. Dropout and L2-Regularization were used to prevent overfitting. Dropout is used in each block’s first and last layer, with a 30% dropout rate, while employed kernel and bias L2-regularization are applied in each layer of identity and convolutional blocks. Both regularization parameters were set to 0.01. Adam optimizer is used to calculate the gradients, with a learning_rate of the optimizer between 0.001 and $1\times 10^{-6}$ and beta_1, beta_2 values of the optimizer set to 0.09, 0.999.

### 6.3. Results

#### 6.3.1. Offline Results

We split the original dataset into train, validation and test sets according to 70–20–10%. We train the model on training set and we make predictions on validation and test set (single hold-out validation). We concentrate on testing accuracy to tune network hyperparameters (e.g., learning rate regularization, batch size, and number of filters in different layers in out mode) are experimentally chosen in a pre-test. The evaluation results on test set per each class are shown in Table 1, Table 2, Table 3, Table 4, Table 5, Table 6 and Table 7.

Moreover, in further experiments, we used K-Fold (stratified sampling) cross-validation [39] to train the proposed model. K-Fold cross-validation is able to generate out-of-sample prediction for every single element in the entire training set and also helps us to estimate number of epochs to train a our model, especially when we use early stopping. Moreover, K-Fold cross-validation helps us to evaluate the effectiveness of hyperparameters (e.g., adding more hidden layers and activation functions, number of convolution filters, learning rate, regularization and dropout). Our original data set splits into training and a test set, such training set split randomly into K subsets/folds, where K=5 in our case. Here, we are dividing observation into five folds. The proposed model is estimated through k−1 subsets, and the $K_{th}$ subset is used for validation. Such a process repeats until each subset is used as a validation set. We evaluate each trained model by testing its performance in the testing set. The resulting evaluation metric called testing average accuracy over each subset/fold, which is the better estimate out of sample performance, and results are averaged across each fold. Finally, we reported average test accuracy or cross-validated accuracy, which is used to estimate out-of-sample accuracy. All the performance results are reported from Table 7 and Table 8. where Table 5 and Table 8 provide single hold out and five fold average accuracy and Table 6 and Table 7 are evaluation results of Mixed Dataset Section 5.2.6 with and without ELU activation function.

Overall, the average accuracy ranges from 80.09% to 100%, suggesting a good ability of the model in recognizing emotions. However, by evaluating the single emotion classification, it appears that the model is not able to generalize in the different datasets well. For instance, Table 1 demonstrates remarkably smaller accuracy for *Happiness* and *Naturalness* classes (e.g., around 63%) than the other emotions (e.g., higher than 97%). Similarly, we can notice the same drawback for the emotions of *Fear* and *Disgust* in Table 2 and *Anger* in Table 4. We hypothesize that such generalization differences are due to a lack of sufficient data per class because of the imbalance in the datasets. Moreover, the presence of ELU provides huge advantage, and Table 7 provides the evaluation results with only ReLU activation function, which shows the model well stable with all the categories and performance also drastically improved with the presence of the ELU activation function. In accordance with these results, the plot of the learning curves in Figure 7 show only tiny fluctuations in overfitting due to class imbalance and insufficient data.

For completeness, we also show the corresponding confusion matrices in Figure 8 to provide a complete overview of the performance of our model over the classes. It is worth noticing that the difference in dimensions are due to the analyzed classes (e.g., seven emotions or eight emotions) in each dataset, and the results are coherent with our previous observation. Despite these differences among the single performance, as we expected, the results achieved by training our model from a scratch with the Mixed dataset composed of samples from all the datasets (please refer to Section 5.2.6 for further details) reveal a more stable performance. Indeed, Table 6 and Figure 8 display similar accuracies over the classes, resulting in a more robust model.

Finally, to validate our architectural choices, we evaluate the performance of the proposed method with respect to other state-of-the-art approaches by focusing on the trade-off between the recognition rate and complexity of the related model. With this purpose, Table 8, Table 9, Table 10 and Table 11 compare our method with the other ones available in the literature previously tested on the same datasets: CK+, Jaffe, KDEF and TFEID. We have excluded the SASE-FE and MIXED datasets because there are no other works evaluating emotion recognition using them. The first has been created for a different aim (e.g., recognizing fake vs. reliable emotions, please refer to Section 5.2.5 for further details), and the latter has been proposed in this work for the first time.

It is worth noticing that the average accuracies of our models are slightly lesser than the state-of-the-art approaches for the CK+ and JAFFE datasets, but with a significant reduction in the dimension of the network (e.g., reduction in the number of parameters) and hence of the required resources. Differently, it is the case of the KDEF dataset in Table 10 that the faces are well centered in the images and are cropped by the background and hair. We hypothesize that this aspect helps the model to easily recognize the emotions and reduce the number of false positives due to artifacts. Moreover, where our tiny models outperforms the other approaches despite the limited number of parameters involved, while Table 11 reveals comparable performance to the other state-of-the-art methods on the TFEID dataset.

#### 6.3.2. Real-Time Results

Finally, the LEMON model has been successfully integrated to process the images from the robot’s camera. We have involved a healthy 28-years-old female who agreed to participate in this study and signed a consent form. We have carried out a pilot experiment in accordance with the principles of the Declaration of Helsinki to assess the feasibility and the reactivity of our model in real time. Specifically, she was asked to reproduce the examined emotions (one per time), while the robot’s camera images and the detected emotions were recorded. A total of 3400 images were collected (average image rate about 5 Hz). The results reveled a prediction time (e.g., the time required by the model to process the input image and classify the current emotion) of 164.76 ± 28.63 ms. The entire distribution of the prediction time is shown in Figure 9.

Overall, the results are consistent with the ones reported in the previous section, showing the feasibility of our model to run in real time also on a commercial robot with limited computational capability. 

## 7. Conclusions and Future Works

In this paper, we have proposed an Emotion recognition system based on Deep Learning and Computer Vision for limited memory devices such as a commercial robot. With this purpose, we have explored how to optimize the network structure underlying the Emotion recognition model in order to limit the computational cost. Specifically, we have exploited a *Residual-learning-based* technique with the combination of *Dilated Convolutional layers* and *Standard 2D Convolutional layers*.

Residual layers accelerate the learning procedure via skip connections, and dilated convolutional layers expand receptive field exponentially without loss of resolution or coverage, and these combinations extensively reduce the computational cost without compromising the performance. Moreover, we included the Exponential Linear Unit to reduce the dying neuron problem from Rectified Linear Unit.

The proposed model has been evaluated with four benchmark datasets and two complementary datasets. The results have demonstrated the robustness, the stability and the ability of the model to work with differences such as scale, background, gender, age and race in the different datasets. Nevertheless, the model might be affected by the presence of class imbalance in the data. In addition, in certain cases (e.g., for the dataset CK+ and JAFFE), the performance appear sub-optimal with respect to the state-of-the-art but with a significant reduction in the size of the network (e.g., number of parameters). However, overall, the performances are quite good (e.g., they range from 80.09% to 100%), and the model is as tiny as possible to run on the real commercial robot (i.e., Ohmni robot).

Future work will include the integration of the proposed Emotion recognition model with other services (e.g., dialogue, navigation, etc.) on the robot in order to personalize the interaction according to the user’s emotional status. In addition, we will focus on producing a new hybrid model based on the emotion recognition from both the robot’s camera images and voice of the people.

## Figures and Tables

**Figure 1 sensors-22-03366-f001:**
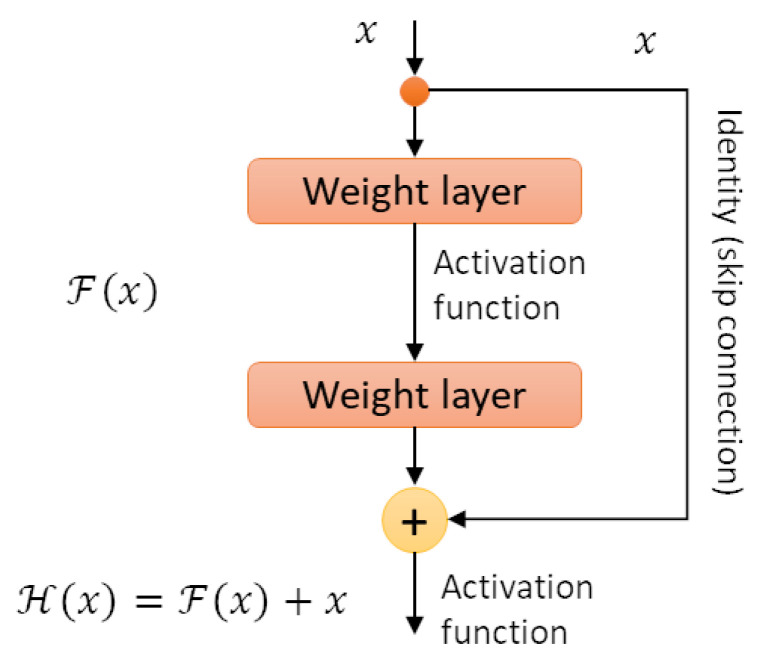
Residual Learning: a building block [27].

**Figure 2 sensors-22-03366-f002:**
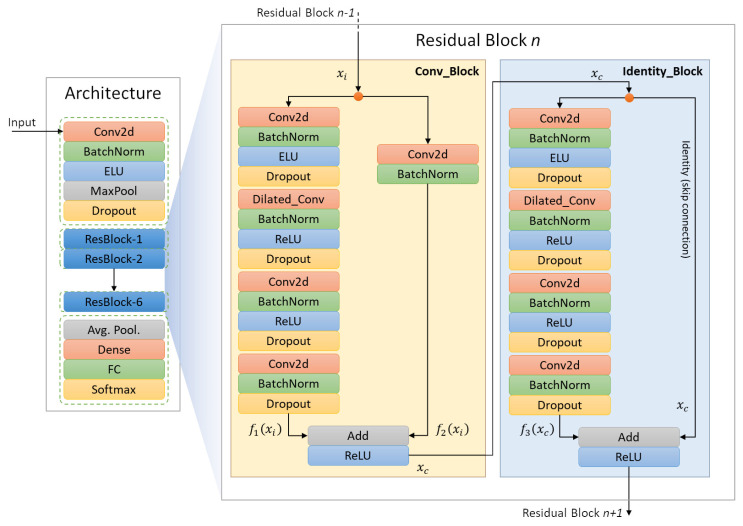
The LEMON Architecture pipeline and a representation of the residual bock including Convolution (**left**) and Identity Block (**right**).

**Figure 3 sensors-22-03366-f003:**
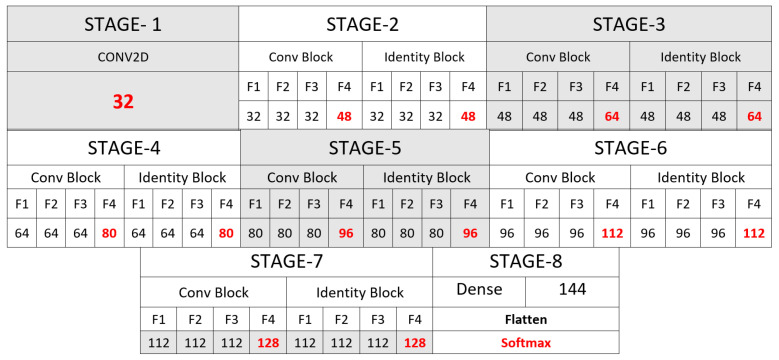
Filter Configuration of LEMON Network. The red values represent the number of filters in the final layer of each block.

**Figure 4 sensors-22-03366-f004:**
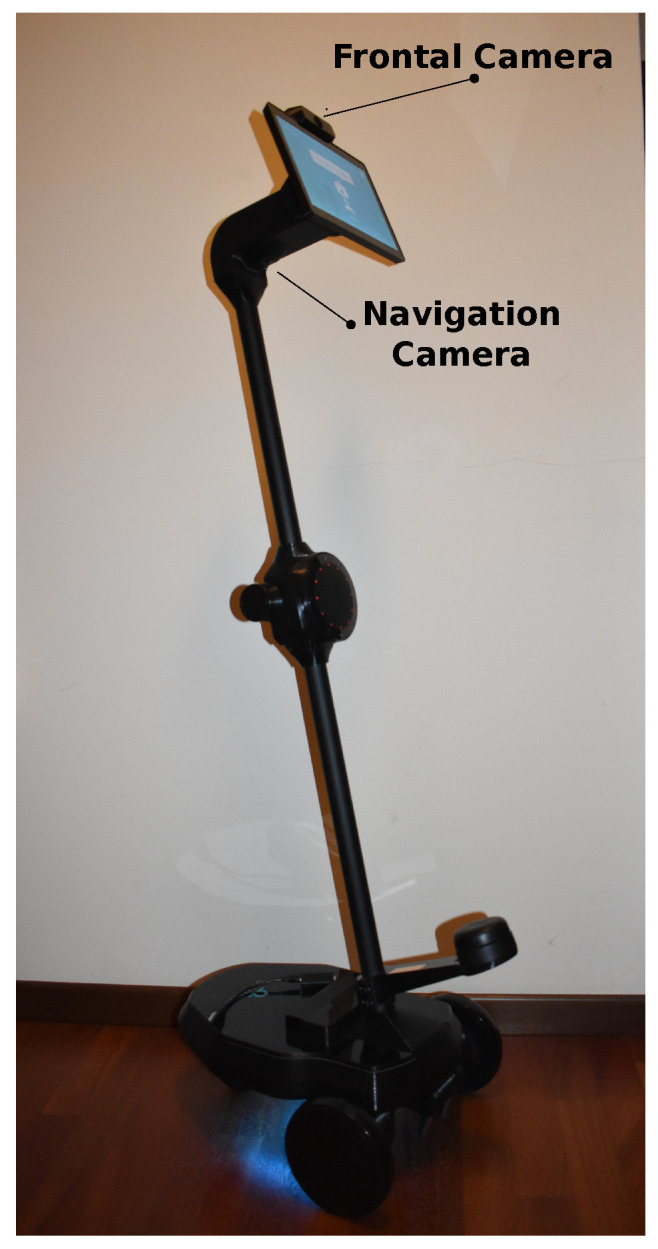
Our robotic platform.

**Figure 5 sensors-22-03366-f005:**
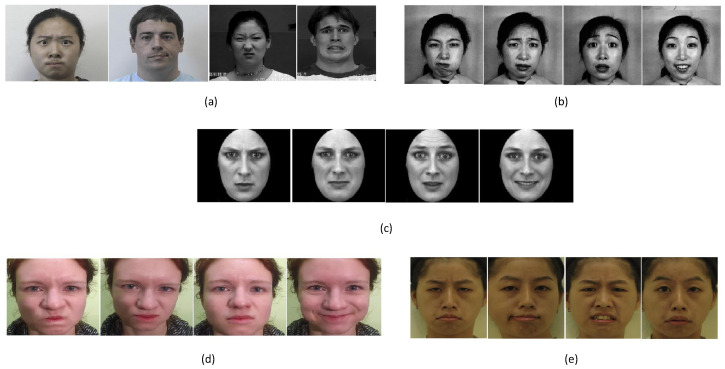
Sample images from (**a**) CK+, (**b**) Jaffe, (**c**) KDEF, (**d**) Sase-FE and (**e**) TFEID Datasets.

**Figure 6 sensors-22-03366-f006:**
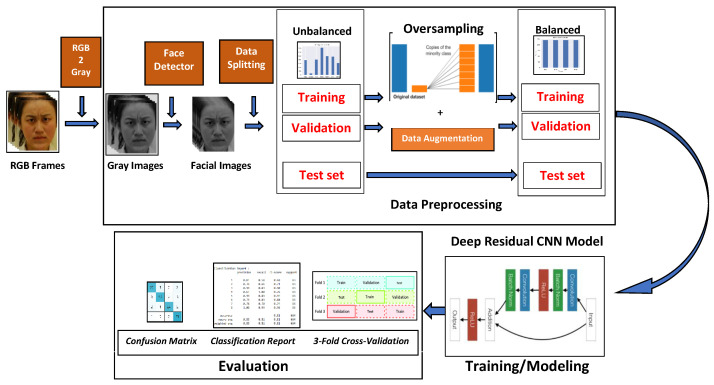
Pipeline for Training and Evaluation: Including Preprocessing, Data Partition, Data Augmentation, Training and Evaluation.

**Figure 7 sensors-22-03366-f007:**
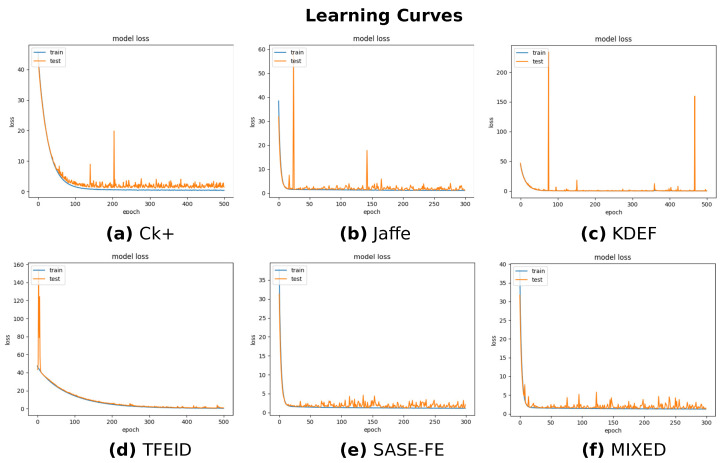
Learning Curves (Loss) over the examined datasets.

**Figure 8 sensors-22-03366-f008:**
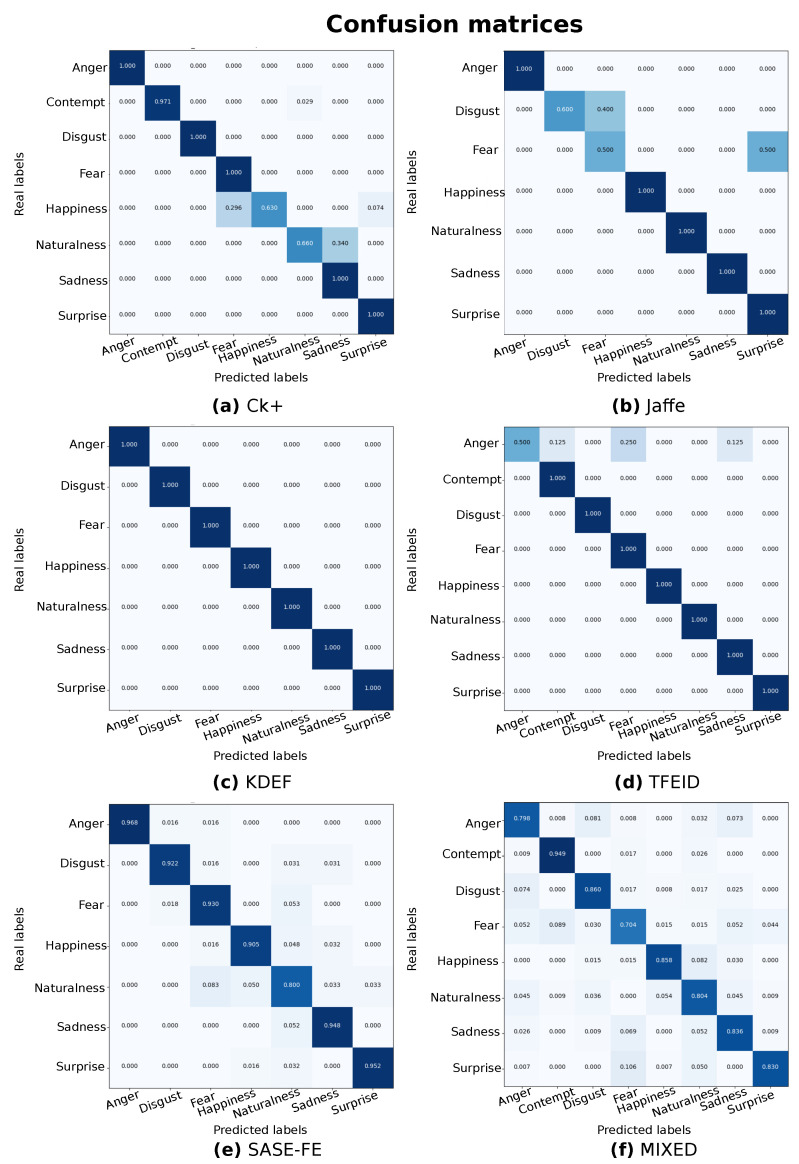
Confusion matrices over the examined datasets. The shades of colors indicate the classification performance. Darker colors are associated with higher prediction accuracy.

**Figure 9 sensors-22-03366-f009:**
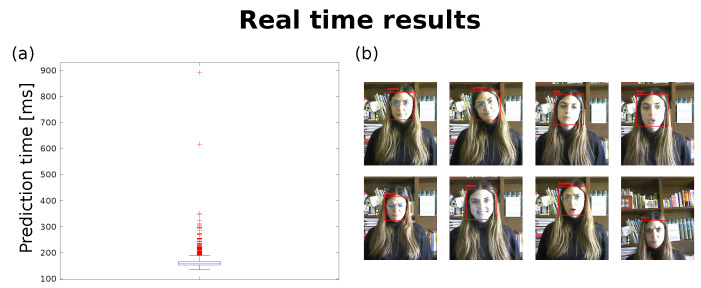
The real-time results: (**a**) The entire distribution of the prediction time; (**b**) Illustrative examples of the detected emotions.

**Table 1 sensors-22-03366-t001:** Performance per class and average accuracy using both models, achieved with one-fold and five-fold cross validation, on test set from CK+ dataset.

Emotion	# Images	Avg. Acc._F1	Avg. Acc._F5
**Anger**	34	100.0%	74.33%
**Contempt**	34	97.1%	93.06%
**Disgust**	34	100.0%	96.47%
**Fear**	34	100.0%	80.0%
**Happiness**	34	63.0%	69.79%
**Naturalness**	34	66.0%	68.61%
**Sadness**	34	100.0%	92.67%
**Surprise**	34	100.0%	99.25%
**Mean**	-	90.76%	84.27%

**Table 2 sensors-22-03366-t002:** Performance per class and average accuracy using both models, achieved with one-fold and five-fold cross-validation, on test set from JAFFE dataset.

Emotion	# Images	Avg. Acc._F1	Avg. Acc._F5
**Anger**	3	100.0%	100.0%
**Disgust**	3	60.0%	56.6%
**Fear**	3	50.0%	53.3%
**Happiness**	3	100.0%	95.0%
**Naturalness**	3	100.0%	82.0%
**Sadness**	3	100.0%	73.66%
**Surprise**	3	100.0%	100%
**Mean**	-	87.14%	80.09%

**Table 3 sensors-22-03366-t003:** Performance per class and average accuracy using both models, achieved with one-fold and five-fold cross-validation, on test set from KDEF dataset.

Emotion	# Images	Avg. Acc._F1	Avg. Acc._F5
**Anger**	29	100.0%	100.0%
**Disgust**	29	100.0%	100.0%
**Fear**	29	100.0%	99.33%
**Happiness**	29	100.0%	100.0%
**Naturalness**	29	100.0%	100.0%
**Sadness**	29	100.0%	100.0%
**Surprise**	29	100.0%	100.0%
**Mean**	-	100.0%	99.90%

**Table 4 sensors-22-03366-t004:** Performance per class and average accuracy using both models, achieved with one-fold and five-fold cross-validation, on test set from TFEID dataset.

Emotion	# Images	Avg. Acc._F1	Avg. Acc._F5
**Anger**	4	50.0%	60.0%
**Contempt**	4	100.0%	76.57%
**Disgust**	4	100.0%	100.0%
**Fear**	4	100.0%	86.0%
**Happiness**	4	100.0%	92.0%
**Naturalness**	4	100%	93.75%
**Sadness**	4	100.0%	76.0%
**Surprise**	4	100.0%	100%
**Mean**	-	93.75%	85.54%

**Table 5 sensors-22-03366-t005:** Performance per class and average accuracy using both models, achieved with one-fold and five-fold cross-validation, on test set from SASE-FE dataset.

Emotion	# Images	Avg. Acc._F1	Avg. Acc._F5
**Anger**	61	96.8%	92.67%
**Contempt**	61	92.2%	92.40%
**Disgust**	61	93%	88.39%
**Happiness**	61	90.5%	89.0%9
**Naturalness**	61	80%	84.6%
**Sadness**	61	94.8%	86.63%
**Surprise**	61	95.2%	95.60%
**Mean**	-	91.78%	89.91%

**Table 6 sensors-22-03366-t006:** Performance per class and average accuracy using both models, achieved with one-fold and five-fold cross-validation, on validation and test set from MIXED dataset with ELU activation function.

Emotion	#Images	Val Avg. Acc._F1	Val. Avg. Acc._F5	#Images	Test Avg. Acc._F1	Test. Avg. Acc._F5
**Anger**	314	81.2%	84.01%	125	79.8%	80.32%
**Contempt**	314	91.2%	90.29%	125	94.9%	89.01%
**Disgust**	314	78.3%	81.75%	125	86.0%	81.74%
**Fear**	314	80.6%	76.95%	125	70.4%	69.81%
**Happiness**	314	85.9%	86.93%	125	85.8%	87.45%
**Naturalness**	314	81.8%	76.71%	125	80.4%	79.13%
**Sadness**	314	82.9%	81.53%	125	83.6%	78.97%
**Surprise**	314	94.8%	90.86%	125	83.0%	86.46%
**Mean**	-	84.58%	83.63%	-	82.98%	81.61%

**Table 7 sensors-22-03366-t007:** Performance per class and average accuracy using both models, achieved with one-fold and five-fold cross-validation, on validation and test set from MIXED dataset without ELU Activation Function.

Emotion	#Images	Val Avg. Acc._F1	Val. Avg. Acc._F5	#Images	Test Avg. Acc._F1	Test. Avg. Acc._F5
**Anger**	314	65.41%	59.42%	125	55.73%	57.61%
**Contempt**	314	21.49%	21.76%	125	21.23%	20.57%
**Disgust**	314	67.71%	63.21%	125	65.30%	62.43%
**Fear**	314	59.90%	61.12%	125	54.54%	56.21%
**Happiness**	314	80.26%	87.66%	125	79.32%	87.45%
**Naturalness**	314	45.0%	55.49%	125	40%	45.76%
**Sadness**	314	61.40%	59.53%	125	52.63%	65.37%
**Surprise**	314	87.28%	84.81%	125	95.65%	89.62%
**Mean**	-	60.94%	61.64%	-	58.01%	60.63%

**Table 8 sensors-22-03366-t008:** Comparison of our approach, trained with one-fold and five-fold cross-validation, with respect to the state-of-the-art methods on the CK+ Dataset in terms of average accuracy and number of parameters.

Item/Method	[15]	[40]	[41]	LEMON_1F	LEMON_5F
Avg. Acc.	93.2%	99.44%	94.9%	90.76	84.27%
# parameters	>23M	61M	>34M	1.6M	1.6M

**Table 9 sensors-22-03366-t009:** Comparison of our approach, trained with one-fold and five-fold cross-validation, with respect to the state-of-the-art methods on the JAFFE Dataset in terms of average accuracy and number of parameters.

Item/Method	[42]	[43]	[44]	LEMON_1F	LEMON_5F
Avg. Acc.	96.44%	92.4%	98.57%	87.14%	80.09%
# parameters	>100 M	26 M	>13 M	1.6 M	1.6 M

**Table 10 sensors-22-03366-t010:** Comparison of our approach, trained with one-fold and five-fold cross-validation, with respect to the state-of-the-art methods on the KDEF Dataset in terms of average accuracy and number of parameters.

Item/Method	[45]	[46]	[47]	LEMON_1F	LEMON_5F
Avg. Acc.	85.90%	97.2%	72.55%	100%	99.90%
# parameters	>24 M	>20 M	138 M	1.6 M	1.6 M

**Table 11 sensors-22-03366-t011:** Comparison of our approach, trained with one-fold and five-fold cross-validation, with respect to the state-of-the-art methods on the TFEID Dataset in terms of average accuracy and number of parameters.

Item/Method	[48]	[49]	[50]	LEMON_1F	LEMON_5F
Avg. Acc.	92.8	93.36	92.54	93.75%	85.82%
# parameters	not reported	>25 M	not reported	1.6 M	1.6 M

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
