# Peer review of "LEMON: A Lightweight Facial Emotion Recognition System for Assistive Robotics Based on Dilated Residual Convolutional Neural Networks"

_sensors, 2022, doi:10.3390/s22093366_

Round 1

Reviewer 1 Report

The proposed LEMON algorithm shows less computation and memory cost to recognize facial expressions. The key suggestion is that the use of ELU is applied to solve the ReLU problem.

1. The residual learning block to be compared is shown in Fig.1. Suggested content compared with Fig.2, the expression is too simplistic. For comparison with the proposed content and structure, a similar level of explanation and flow block expression are needed.

2.  In Fig. 2, the advantage of skip connection part is not explained. It is necessary to express and explain the skip connection behavior in each of the eight building blocks.

3. It is necessary to explain why the LEMON method  shows exceptionally high ACC in the KDEF dataset in Tables 7, 8, and 9.

4. It was said that the proposed method has strengths in computation and memory cost. Expression prediction time and real-time verification are required to verify computation performance.

5. minor points : Correct the repetition of "data" on line 117, and modify "N"-->"Y", "y"-->"j" in Equation (1) on line 157. Replace "methodologies" with "datasets" on line 252.

Author Response

Dear Editors,

please find enclosed the revised version of the paper:

LEMON: A Lightweight Facial Emotion Recognition System for Assistive Robotics based on Dilated Residual Convolutional Neural Networks

by Rami Reddy Devaram, Gloria Beraldo,  Riccardo De Benedictis, Misael Mongiovì, Amedeo Cesta

Submitted to Sensors with reference number sensors-1668584

We would like to thank all the reviewers for their useful and valuable comments. We revised our paper according to their comments aiming at creating a new version of our article and responding to all the required changes. New added/modified parts of text are now highlighted in blue. In the following, we summarize the changes we made on the paper subdivided according to the reviewers’ requests.

Best regards,

Rami Reddy Devaram, Gloria Beraldo,  Riccardo De Benedictis, Misael Mongiovì, Amedeo Cesta.

Comment 1.1

The residual learning block to be compared is shown in Fig.1. Suggested content compared with Fig.2, the expression is too simplistic. For comparison with the proposed content and structure, a similar level of explanation and flow block expression are needed.

Answer 1.1

We thank the reviewer for this comment. Figure 1 represents the building block of a residual network. We have included such a building block in our architecture which is depicted in Figure 2. Specifically, we better explained the residual learning in Section 3.3 and how Figure 1 and Figure 2 are correlated with each other, trying to adopt the same formalism to facilitate the comparison, in Section 4.

Comment 1.2

In Fig. 2, the advantage of skip connection part is not explained. It is necessary to express and explain the skip connection behavior in each of the eight building blocks.

Answer 1.2

We thank the reviewer for this comment. Skip connections are standard modules in various deep neural network architectures. They provide an alternative path for the gradient, which are often beneficial for the model convergence by skipping some layers in the neural network. We have added some paragraphs in Section 3.3 and in Section 4 to better explain skip connections and how we used them in our architecture.

Comment 1.3

It is necessary to explain why the LEMON method  shows exceptionally high ACC in the KDEF dataset in Tables 7, 8, and 9.

Answer 1.3

We thank the reviewer for this comment. In the KDEF dataset the faces are well centered in the images and are cropped by the background and hair. We hypothesize that this aspect facilitates the model to easily recognize the emotions and reduce the number of false positives due to artifacts.  We have explained these details in Section 6.3 providing some hints about such a high accuracy.

Comment 1.4

It was said that the proposed method has strengths in computation and memory cost. Expression prediction time and real-time verification are required to verify computation performance.

Answer 1.4

We thank the reviewer for this precious comment. According to the reviewer’s suggestion, in the revised version of the paper, we have verified the feasibility of the proposed model by performing a pilot experiment in real time with one participant. The results revealed a prediction time of 164.76 ± 28.63 ms (see Figure 9) and an average accuracy of XX. Overall, the real-time results are in line with the offline ones.

Further details about this pilot test are reported in the Section 6.3.2. that we added in the revised manuscript.

Comment 1.5

minor points : Correct the repetition of "data" on line 117, and modify "N"-->"Y", "y"-->"j" in Equation (1) on line 157. Replace "methodologies" with "datasets" on line 252.

Answer 1.5

We thank the reviewer for this helpful comment. All typos have been corrected.

Reviewer 2 Report

This paper focuses on facial emotion recognition. The authors proposed a method named LEMON and proved the effectiveness by experiment on small datasets. Before acceptance, there are several problems to be addressed:

(1) The presentation of the technical part in Section 4 should be improved for readers to understand.

(2) How to determine the train and test set images? In Table 2 and Table 4, is 3 or 4 images enough for tests?

(3) Figure 3 is not aligned properly.

(4) Please check the positions of the superscript and the subscript of Equation (1), such as the position of  l=1 and X?

(5) More related works of DCNN may help to improve the paper, such as 
An efficient deep learning technique for facial emotion recognition, MTA, 2022;
Deep embedding of concept ontology for hierarchical fashion recognition,  Neurocomputing, 2021;
Distilling Ordinal Relation and Dark Knowledge for Facial Age Estimation, TNNLS, 2021.

Author Response

Dear Editors,

please find enclosed the revised version of the paper:

LEMON: A Lightweight Facial Emotion Recognition System for Assistive Robotics based on Dilated Residual Convolutional Neural Networks

by Rami Reddy Devaram, Gloria Beraldo,  Riccardo De Benedictis, Misael Mongiovì, Amedeo Cesta

Submitted to Sensors with reference number sensors-1668584

We would like to thank all the reviewers for their useful and valuable comments. We revised our paper according to their comments aiming at creating a new version of our article and responding to all the required changes. New added/modified parts of text are now highlighted in blue. In the following, we summarize the changes we made on the paper subdivided according to the reviewers’ requests.

Best regards,

Rami Reddy Devaram, Gloria Beraldo,  Riccardo De Benedictis, Misael Mongiovì, Amedeo Cesta.

Comment 2.1

The presentation of the technical part in Section 4 should be improved for readers to understand.

Answer 2.1

We thank the reviewer for this comment. As per the suggestion, we have added some paragraphs in Section 4 highlighting better the working principle about standard residual learning and how it is related to our proposed architecture, hence explaining the reasons behind such significant performance improvements. We have also reworded some paragraphs in Section 4 to make them more readable.

Comment 2.2

How to determine the train and test set images? In Table 2 and Table 4, is 3 or 4 images enough for tests?

Answer 2.2

We thank the reviewer for this comment. The dataset is divided into train, validation and test sets according to 70%, 20% and 10% respectively (as we reported in Section 6.1). However, some datasets have very few samples (e.g., 24 total images for the JAFFE dataset, and 32 total images for the TFEID dataset). Specifically, the 10% of total images corresponds to 3 samples per class in the JAFFE dataset (results in Table 2) and four per class in TFEID dataset (results in Table 4).

Comment 2.3

Figure 3 is not aligned properly.

Answer 2.3

We thank the reviewer for this comment. We have aligned Figure 3 in the revised manuscript and we take the opportunity to also better center Figure 4.

Comment 2.4

Please check the positions of the superscript and the subscript of Equation (1), such as the position of  l=1 and X?

Answer 2.4

We thank the reviewer for this comment. We have updated Equation (1)  as suggested.

Comment 2.5

More related works of DCNN may help to improve the paper, such as

An efficient deep learning technique for facial emotion recognition, MTA, 2022;

Deep embedding of concept ontology for hierarchical fashion recognition,  Neurocomputing, 2021;

Distilling Ordinal Relation and Dark Knowledge for Facial Age Estimation, TNNLS, 2021.

Answer 2.5

We thank the reviewer for this comment. The suggested references are well related to our work. We have added them in Section 2.

Reviewer 3 Report

The paper proposed a facial expression recognition model that contains a small number of parameters (compared to existing work)

Overall the purpose of the study has merit but the writing style and the motivation on small models for the problem shall be improved. The comments are the following.

-The connection between each paragraph in section 2 is not clear.

- The explanation of what is X in Figure 1 shall be given.

- The content in each part is not cohesive. For example, at the end of 3.3,  the authors should give an overview of residual architecture before referring to section 4 as there are sections 3.4 and 3.5 in between.

- The comparison between RELU and ELU shall be conducted experimentally, in addition, to analytically. (The previous research that shown that ELU is superior is on different input types (NLP vs image).

- The evaluation protocol of the references shall be discussed. (The authors mentioned 1-fold and 5-fold cross-validation for the proposed model but did not mention the references). In addition, the usage of n-fold cross-validation shall be explained in detail, e.g. which hyperparameters are adjusted.

- The result demonstrates that the model contains less number of parameters and that comes at the cost of performance degradation. The authors should make strong motivation on why smaller models are important on target devices, e.g., demonstrates the runtime and percentages of memory usages on target devices.

Author Response

Dear Editors,

please find enclosed the revised version of the paper:

LEMON: A Lightweight Facial Emotion Recognition System for Assistive Robotics based on Dilated Residual Convolutional Neural Networks

by Rami Reddy Devaram, Gloria Beraldo,  Riccardo De Benedictis, Misael Mongiovì, Amedeo Cesta

Submitted to Sensors with reference number sensors-1668584

We would like to thank all the reviewers for their useful and valuable comments. We revised our paper according to their comments aiming at creating a new version of our article and responding to all the required changes. New added/modified parts of text are now highlighted in blue. In the following, we summarize the changes we made on the paper subdivided according to the reviewers’ requests.

Best regards,

Rami Reddy Devaram, Gloria Beraldo,  Riccardo De Benedictis, Misael Mongiovì, Amedeo Cesta.

Comment 3.1

The connection between each paragraph in section 2 is not clear.

Answer 3.1

We thank the reviewer for this comment. We have extended Section 2 with some paragraphs to better connect the existing paragraphs and to better explain the related work.

Comment 3.2

The explanation of what is X in Figure 1 shall be given.

Answer 3.2

We thank the reviewer for this comment. We have improved the description of Figure 1 by highlighting the skip connection and their role in providing an alternative path for the gradient. We have also better explained the formalism adopted in the Figure (e.g. X, F(X), X_identity, etc.).

Comment 3.3

The content in each part is not cohesive. For example, at the end of 3.3,  the authors should give an overview of residual architecture before referring to section 4 as there are sections 3.4 and 3.5 in between.

Answer 3.3

We thank the reviewer for this comment. In the revised manuscript, we have added a brief description on the top of Section 3.3 to better clarify the purpose of the section and the link with Section 4. Furthermore, we have included more details of the residual architecture in Section 3.3 to facilitate the comprehension of the state-of-the-art concepts on which the proposed model is based on.

Comment 3.4

The comparison between RELU and ELU shall be conducted experimentally, in addition, analytically. (The previous research that showed that ELU is superior is on different input types (NLP vs image).

Answer 3.4

We thank the reviewer for this comment. We conducted the required experiment and we reported, in Table 7, the results using only the RELU activation function. By comparing them with the ones shown in Table 6 (ELU + RELU as activation functions), it is worth noticing that the resulting model is more stable in all the classes and performs drastically better with the presence of ELU activation function. Furthermore, we better analyze the advantage provided by ELU activation function analytically in Section 4 by reporting other previous studies exploiting images as input.

Comment 3.5

The evaluation protocol of the references shall be discussed. (The authors mentioned 1-fold and 5-fold cross-validation for the proposed model but did not mention the references). In addition, the usage of n-fold cross-validation shall be explained in detail, e.g. which hyperparameters are adjusted.

Answer 3.5

We thank the reviewer for this comment. In the revised version of the paper, we have better explained the evaluation protocol in Section 6.3.1. by adding references related to 1-fold and 5-fold cross validation and the setting of hyperparameters in our pipeline.

Comment 3.6

The result demonstrates that the model contains less number of parameters and that comes at the cost of performance degradation. The authors should make strong motivation on why smaller models are important on target devices, e.g., demonstrates the runtime and percentages of memory usages on target devices.

Answer 3.6

We thank the reviewer for this helpful comment. Since the required memory and the prediction time are proportional to the number of parameters, we are interested in having smaller models which, however, may have lower accuracy due to the reduced number of parameters. We have added some sentences in the introduction to further motivate our choices. According to your suggestion, we have conducted a pilot experiment in real time to evaluate the feasibility of detecting emotions based on the stream of images from the robot's camera. The results are reported in the Section 6.3.2 in terms of prediction time.

Round 2

Reviewer 3 Report

The authors have done well in addressing all the comments.